# Federated Fairness without Access to Demographics

**Afroditi Papadaki**
University College London
a.papadaki.17@ucl.ac.uk

**Natalia Martinez**
IBM Research
natalia.martinez.gil@ibm.com

**Martin Bertran**
Amazon Web Services
maberlop@amazon.com

**Guillermo Sapiro**
Duke University; Apple
guillermo.sapiro@duke.edu

**Miguel Rodrigues**
University College London
m.rodrigues@ucl.ac.uk

## Abstract

Existing federated learning approaches address demographic group fairness assuming that clients are aware of the sensitive groups. Such approaches are not applicable in settings where sensitive groups are unidentified or unavailable. In this paper, we address this limitation by focusing on federated learning settings of fairness without demographics. We present a novel objective that allows trade-offs between (worst-case) group fairness and average utility performance through a hyper-parameter and a group size constraint. We show that the proposed objective recovers existing approaches as special cases and then provide an algorithm to efficiently solve the proposed optimization problem. We experimentally showcase the different solutions that can be achieved by our proposed approach and compare it against baselines on various standard datasets.

## 1  Introduction

Federated learning (FL) enables different entities to collaboratively learn a statistical model through an iterative procedure that is coordinated by a central server in a decentralized manner. The clients keep their raw data locally to preserve privacy, and share only focused updates with the server, which limits the information shared with the server while still achieving some global learning objective [11, 12]. These learning setups are characterized by data heterogeneity and unbalancedness across participants [10, 25].

A key challenge in federated learning is fairness. Existing literature is predominately focused on group fairness scenarios, where the goal is to learn a model that performs well across pre-defined sensitive groups.[1] The definition of a group varies across different federated learning approaches. The majority of these works [3, 9, 14, 18, 21, 27] characterize the clients as the collection of groups in the federation and propose methods to control performance disparities across them. Others [2, 28] suggest that groups are associated to the available within-client populations and provide methods to tackle within-client group disparities. More recently, the use of global demographics was proposed in [8, 19] for guaranteeing fairness across any sensitive groups included in the federation, thereby enabling clients that have access only to a subset of the required groups to benefit from the training procedure.

---

[1]We use the terms (protected/sensitive/demographic) group/population, interchangeably.

Workshop on Federated Learning: Recent Advances and New Challenges, in Conjunction with NeurIPS 2022 (FL-NeurIPS'22). This workshop does not have official proceedings and this paper is non-archival.

The aforementioned approaches assume that the participants have knowledge about the sensitive demographic groups during training time and that the group memberships have been correctly assigned to each data point. Unfortunately, the assumption that every client has access to predefined and accurate sensitive groups might be unrealistic for various federated learning scenarios. For example, consider a scenario where multiple medical institutions collaborate to learn a model to accurately perform a particular medical diagnosis that is fair across individuals of different demographic (sub)groups. Each institution owns a database consisting of patient data that cannot directly share with other participants due to privacy regulations (e.g. GDPR [1]). Furthermore, the documents facilitated at each local database might lack particular information (e.g. race, religion, sexual orientation, etc.) that the patients did not wish to disclose. The absence of such information during the training process makes existing FL fairness methods, such as [2, 8, 19], inappropriate for learning a model that is fair to patients of different demographics. We illustrate this example in Figure 1.

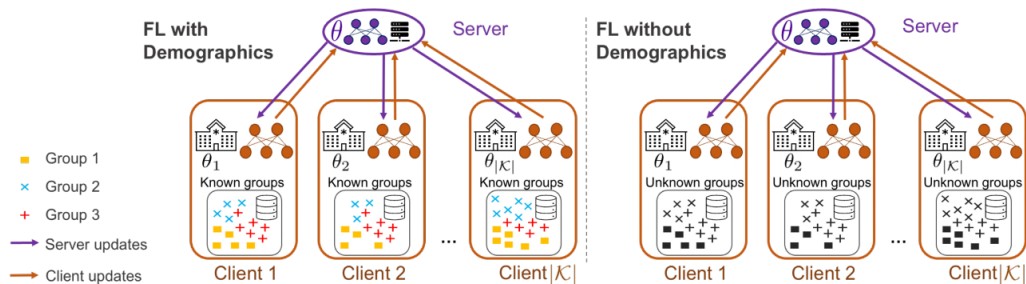

Figure 1: (*Left*) Each hospital $k \in \mathcal{K}$ has prior knowledge about the sensitive populations in its dataset and the global model $h$, parametrized by a vector $\boldsymbol{\theta}$, is trained by incorporating the information about demographics in the training procedure using existing FL fairness methods [2, 8, 19]. (*Right*) Clients are unaware of the local demographics and existing approaches cannot be deployed.

In this paper, we address the problem of group fairness in federated learning scenarios where the clients are unaware about the demographic group composition in their data. We propose a flexible optimization objective that admits a family of solutions that depend on a single hyperparameter $\epsilon$. For $\epsilon = 1$ our objective reduces to the standard ERM problem which does not cater for group fairness. For $\epsilon \approx 0$ it recovers an optimization problem that seeks to minimize the worst-tail risk, subject to a probability-level constraint which is predefined based on some common set of policies and/or preferences across clients, that guarantees subgroup robustness and good performance on unidentified groups.

Our approach helps the participants (a) identify their global vulnerable (testing) population, even if it does not exist on its local distribution during training time and (b) learn a global hypothesis that allows a trade-off between average performance and subgroup robustness. We also present links to popular robustness formulations in centralized machine learning and provide an algorithm that solves our proposed objective. We empirically study the different trade-offs that can be achieved by the proposed approach and experimentally demonstrate the efficiency of the proposed method against other baselines in centralized and federated learning settings.

## 2 Related Work

**Fairness without Demographics in ML.** There is limited research in centralized machine learning trying to address group fairness without explicit demographics. One way to deal with data when group labels are not known is through proxy fairness [6]. Such approaches [6, 24, 29] develop a new proxy variable to substitute the true sensitive group variable so that conventional group fairness methods can be deployed. However, these approaches require the true group variable to be known while the samples group label are considered unavailable, which is not feasible for many applications.

An alternative line of work that addresses group fairness when demographics are completely unknown is (sub)group robustness [7, 13, 15]. These works aim to improve the performance on any possible sensitive population that can be generated that exceeds a predefined size. For example, the authors in [7] aim to minimize the worst-case risk that exceeds a high-risk threshold. Similarly, blind Pareto fairness (BPF) [15] aims to minimize the worst case risk that induced by the worst possible group

distribution, subject to a sufficient group size, while ensuring that the produced solution is also Pareto optimal [17]. Our work departs from these methods since they require the data to be aggregated in a single entity that can be accessed anytime, while we focus on distributed settings where data is hosted in different entities and cannot be shared or accessed directly. Building upon these prior works, we propose a relaxed superquantile formulation that is flexible, in the sense that, it allows achieving different levels of (minimax) group fairness through a hyperparameter $\epsilon$.

**Fair Federated Learning.** There are various fairness definitions in the federated learning literature. A class of popular approaches [3, 9, 18, 27] focuses on ensuring client fairness – or equivalently client-robustness – by learning a model that optimizes for the worst performing client (or cluster of clients). Nevertheless, as formally shown in [19] fairness across clients does not necessarily guarantee fairness across different demographic groups except if the client owns data from a single demographic population.

Several recent works [2, 28] consider group fairness to be a unique objective for each client that is associated only to the groups available in each client, while others [8, 19, 23] propose frameworks for learning group fair models independently of the demographics available in each client. Similar to [2, 8, 19, 23, 28], our goal is to learn a model that ensures (demographic) group fairness across any groups that exist in the clients data. The key difference though is that we explicitly focus on scenarios where the local sensitive populations are completely unknown to the clients during the training.

## 3 Problem Formulation

### 3.1 Preliminaries

Let the pair of random variables $(X, Y) \in \mathcal{X} \times \mathcal{Y}$ represent the input features, and targets, generated from a distribution $p(X, Y)$. Let also a convex loss function $\ell : \Delta^{|\mathcal{Y}|-1} \times \Delta^{|\mathcal{Y}|-1} \to \mathbb{R}_+$ and a convex hypothesis $h$ drawn from a hypothesis class $\mathcal{H} = \{h : \mathcal{X} \to \Delta^{|\mathcal{Y}|-1}\}$, where $\Delta^{|\mathcal{Y}|-1}$ is the probability simplex of dimension $|\mathcal{Y}| - 1$. In this work we consider scenarios where there is not any prior knowledge about the underlying group associated with a particular pair of features-target at training time. We can address this challenge by optimizing the expected risk of the input features-target pairs that exceed a certain threshold, akin to [7, 15]. In particular, let $L_{X,Y} := \ell(h(X), Y)$ denote a random variable representing the loss associated with a hypothesis $h \in \mathcal{H}$. For some predefined probability $\rho \in (0, 1)$, the $(1 - \rho)$-quantile function is defined as

$$q_{L_{X,Y}}(1 - \rho) := \inf \left\{ \beta \in \mathbb{R} : p(L_{X,Y} \leq \beta) \geq 1 - \rho \right\}, \tag{1}$$

and the $(1 - \rho)$-superquantile[2] function at confidence level $(1 - \rho)$ is defined as

$$CVaR_{(1-\rho)}(L_{X,Y}) = \underset{(X,Y)\sim p(X,Y)}{\mathbb{E}}[L_{X,Y}|L_{X,Y} \geq q_{L_{X,Y}}(1 - \rho)]. \tag{2}$$

We note that Eq. 2 is a measure of the upper tail behavior of the distribution $p(L_{X,Y})$ and, as shown in [22], it can be expressed as a (variational) optimization problem

$$CVaR_{(1-\rho)}(L_{X,Y}) = \min_{c\in\mathbb{R}} \left\{ c + \frac{1}{\rho} \underset{(X,Y)\sim p(X,Y)}{\mathbb{E}} \left[ (L_{X,Y} - c)_+ \right] \right\}, \tag{3}$$

where $(\cdot)_+ := \max\{0, \cdot\}$ and the second term in the objective represents the regret of any realizations of $L_{X,Y}$ that are positive. Note that the argument that minimizes the objective in Eq. 3 corresponds to the quantile $q_{L_{X,Y}}(1 - \rho)$. We also note that if the selected loss function is bounded, i.e. $0 \leq \ell(h(x), y) \leq B, \forall(x, y) \in \mathcal{X} \times \mathcal{Y}$, we can equivalently optimize over $c \in [0, B]$. If we consider that the unknown demographic groups can have a minimum size $\rho$, following [15, 26] we can formulate the problem of learning a minimax group fair hypothesis $h^*$, as

$$\min_{h\in\mathcal{H},c\in\mathbb{R}} \left\{ c + \frac{1}{\rho} \underset{(X,Y)\sim p(X,Y)}{\mathbb{E}} [(L_{X,Y} - c)_+] \right\} \tag{4}$$

The problem in Eq. 4 can be applied in scenarios where the data are gathered in a single entity and can be accessed directly throughout the training process. We next show how this problem can be modified for federated learning scenarios.

---

[2]We use the terms conditional value at risk (CVaR), expected tail loss, and superquantile, interchangeably.

## 3.2 Federated Fairness with Unknown Groups Setting

In the context of federated learning, we consider an additional random variable $K \in \{1, \ldots, |\mathcal{K}|\}$ which represents the clients participating in the federation. Each client $k \in \mathcal{K}$ holds data modelled by its own local distribution $p(X, Y | K = k) = p(X | K = k)p(Y | X, K = k)$. Therefore, the data of the entire federation can be described with the mixture distribution $p(X, Y) = \sum_{k \in \mathcal{K}} p(K = k)p(X, Y | K = k)$. Let $L_{X,Y|K=k} := \ell(h(X), Y)$, with $(X, Y) \sim p(X, Y | K = k)$, denote a random variable representing the local loss induced by a hypothesis $h$. Then the problem in Eq. 3 can be equivalently expressed for federated learning settings as

$$\min_{h \in \mathcal{H}, c \in \mathbb{R}} CVaR_{(1-\rho)}(L_{X,Y}) = \min_{h \in \mathcal{H}} \mathbb{E}_{K \sim p(K)} \left[ \mathbb{E}_{X,Y \sim p(X,Y|K=k)} [(L_{X,Y|K=k} - c)_+] \right] \quad (5)$$

The $CVaR_{(1-\rho)}$ in RHS of Eq. 5 ignores any data that is not considered high-risk which in turn might unnecessarily reduce the utility performance of the obtained solution on it. Especially, if there are regions in the input space where there is no uncertainty about the target class, i.e., the data is perfectly separable. Motivated by this issue, we propose a relaxed version of Eq. 5, namely R-CVaR, where a hyperparameter $\epsilon \in (0, 1]$ accommodates a compromise between the average performance and worst-samples performance as follows

$$\min_{h \in \mathcal{H}} \left\{ (1 - \epsilon)CVaR_{(1-\rho)}(L_{X,Y}) + \epsilon \mathbb{E}_{(X,Y) \sim p(X,Y)} [L_{X,Y}] \right\} =$$

$$\min_{h \in \mathcal{H} c \in \mathbb{R}} \mathbb{E}_{K \sim p(K)} \left[ (1 - \epsilon) \mathbb{E}_{X,Y|K=k} [(L_{X,Y|K=k} - c)_+] + \epsilon \mathbb{E}_{X,Y|K=k} [L_{X,Y|K=k}] \right] \quad (6)$$

The main benefit of objective in Eq. 6 is that it supports a continuum of solutions that depend on the different values of $\epsilon$. Setting $\epsilon \approx 0$ allows to focus on the worst-performing sample at the cost of the low risk samples. For such small $\epsilon$ values we assert that the hypothesis solving Eq. 6 is minimax properly Pareto optimal [5, 17]. On the other hand, picking $\epsilon = 1$, the proposed objective generalizes prior work in federated learning settings and becomes the vanilla-ERM problem in FedAvg [16]. For any other intermediate value of $\epsilon \in (0, 1)$, we achieve a trade-off between utility[3] and subgroup robustness. To better understand the set of trade-offs achieved by the proposed objective, we offer an illustrative example in Figure 2. We note that the value of $\epsilon$ is predefined and fixed, and therefore, we leave it to up to the policy maker(s) to determine it.

**Remark 1.** *The hypothesis that determines the $(1 - \rho)$-quantile c, for which $c \geq L_{X,Y}$, is the uniform classifier $\bar{h} : \bar{h}_y(X) = \frac{1}{|\mathcal{Y}|} \forall y \in \mathcal{Y}$.*

## 4 Optimization

In real applications, each client holds a finite dataset $D_k = \{(x_i, y_i)\}_{i=1,\ldots,n_k}$ sampled from the true distribution $p(X, Y | K = k)$, with $D = \bigcup_{k \in \mathcal{K}} D_k$ being the dataset containing all the data samples available across clients of size $n = \sum_{k \in \mathcal{K}} n_k$. We explicitly express the empirical form of the proposed R-CVaR objective, as follows

$$\min_{\theta \in \Theta, c \in [0,B]} \sum_{k \in \mathcal{K}} \frac{n_k}{n} \left[ (1 - \epsilon)[c + \frac{1}{n_k \rho} \sum_{i=1}^{n_k} (\ell(x_{i,k}, y_{i,k}; \theta) - c)_+] + \frac{\epsilon}{n_k} \sum_{i=1}^{n_k} \ell(x_{i,k}, y_{i,k}; \theta) \right], \quad (7)$$

where $\theta \in \Theta$ is the vector parametrizing the hypothesis $h \in \mathcal{H}$.

We now introduce an algorithm to solve the empirical objective of Eq. 7, in a federated way. Our algorithm consists of two main steps: (a) model parameters update and (b) periodic calculation of the threshold $c$.

**Model update.** Learning the model is a simple procedure where the participating clients receive the current round model parameters $\theta^t$ from the server, perform an optimization step and return the updated model parameters $\theta_k^t$ to the server. Then, the server produces the new model parameters $\theta^{t+1}$ by averaging the received client model parameters.

---

[3]We define as utility the mean model performance.

**Threshold calculation.** The global model parameters $\theta$ and quantile $c$ can only be computed in the server, since it requires a collection of relevant information from the clients. One key challenge imposed by the proposed objective is the efficient calculation of the quantile $c$ in a federated fashion without vastly increasing the communication overhead per client. Here, we propose a practical technique for estimating this parameter.

Let $c$ be the estimated quantile. We denote $\rho(c) = \sum_{k \in \mathcal{K}} p(K = k)\rho_k(c)$ the estimated probability, with $\rho_k(c) = \mathbb{E}_{(X,Y)\sim p(X,Y|K=k)}[\mathbb{1}(L_{X,Y|K=k} \geq c)]$. Also, by definition, we have that the objective's $\rho = \mathbb{E}_{(X,Y)\sim p(X,Y)}[\mathbb{1}(L_{X,Y} \geq q_{L_{X,Y}}(1 - \rho))]$ is equal to $\rho(c)$. Thus, we can compute the quantile $c$ that satisfies for a fixed $\rho$ by ensuring that the objective's and estimated probabilities at each communication round are the same, i.e. $\rho = \rho(c)$. This can realized by the optimization procedure $\min_c (\rho - \rho(c))^2$. As a result, we can update the estimated threshold $c$ according to

$$c^{t+1} \leftarrow \prod_{c \in [0,B]} \left( c^t - \eta_c \operatorname{sign}(\rho - \rho(c^t)) \right). \tag{8}$$

Note that $\eta_c$ captures the product of the learning rate and the absolute value of the derivative $(\rho - \rho(c))^2$ w.r.t. $c$. We summarize the proposed solver in Algorithm 1.

---

**Algorithm 1** FEDRCVAR ALGORITHM

---

**Inputs:** $\mathcal{K}$: set of clients, $T$: communication rounds, $\eta_{\boldsymbol{\theta}}$: model learning rate, $\eta_c$: learning rate for quantile $c$, $\epsilon \in (0,1)$: trade-off parameter, $\rho \in (0,1)$: parameter for probability-level, $c^0$: initial threshold set to $B$.

1: Server initializes $\boldsymbol{\theta}^0$ randomly.
2: **for** $t = 1$ to $T$ **do**
3:     Server **broadcasts** $\boldsymbol{\theta}^{t-1}$ and $c^t$
4:     **for** each client $k \in \mathcal{K}$ **in parallel do**
5:         $\boldsymbol{\theta}_k^t \leftarrow \boldsymbol{\theta}^{t-1} - \eta_{\boldsymbol{\theta}} \nabla_{\boldsymbol{\theta}} \left\{ \frac{1-\epsilon}{n_k} \sum_{i=1}^{n_k} (\ell(h(x_k^i), y_k^i) - c^t)_+ + \frac{\epsilon}{n_k} \sum_{i=1}^{n_k} \ell(h(x_k^i), y_k^i) \right\}$
6:         Return local model $\boldsymbol{\theta}_k^t$ and $\rho_k(c^t) = \frac{1}{n_k} \sum_{i=1}^{n_k} \mathbb{1}(\ell(h(x_k^i), y_k^i) \geq c^t)$ to server
7:     **end for**
8:     $\boldsymbol{\theta}^t \leftarrow \sum_{k \in \mathcal{K}} \frac{n_k}{n} \boldsymbol{\theta}_k^t$
9:     $c^{t+1} \leftarrow \prod_{c \in [0,B]} \left( c^t - \eta_c \operatorname{sign}(\rho - \sum_{k \in \mathcal{K}} \frac{n_k}{n} \rho_k(c^t)) \right)$
10: **end for**
**Output:** $\boldsymbol{\theta}^T$

---

## 5 Experimental Results

We show the benefits of the proposed approach in a *synthetic dataset* (see A.1 for dataset generation details) and two real world datasets: (a) *eICU dataset* [20] critical care dataset to predict patient mortality using real patient documents from different medical centers. For the FL approaches, we split the data to 11 clients, each representing one of the hospitals in the dataset. (b) *ACS Employment dataset* [4] for employment classification based on 14 input features. In FL settings, we distribute the data to 3 clients so that each client represents a race from {Black, White, Others}. For the centralized settings, there are no clients involved and we use all the available training data during the training process. We provide more information about experimental details in Appendix A.

First, we empirically explore the various levels of group fairness that can be achieved through $\epsilon$, for a given size $\rho$, when we optimize the R-CVaR in Eq.6. This is shown in Figure 2. The R-CVaR objective magnifies the impact of worst performing group and decreases the focus on the average performance for $\epsilon \approx 0$. Setting $\epsilon = 1$ recovers the ERM objective. We also observe that the smaller the value of $\rho$ the larger the trade-offs between fairness and utility we can achieve through $\epsilon$. Note

that when $\epsilon$ is closer to 0, and $\rho$ is sufficiently small the worst-group risk is closer to the uniform classifier risk. This is consistent to the observations made in [15] about the existence of a critical $\rho$.

Next, we compare the proposed approach against centralized ML approaches and relevant FL approaches in Figure 3. We benchmark against (a) BPF which guarantees subgroup robustness, (b) empirical risk minimization (ERM), (c) AFL which ensures client-fairness (or equivalently client robustness), and (d) FedAvg which optimizes for utility in FL settings. We omit comparisons to approaches that require predefined groups. As expected, FedRCVaR admits the Pareto optimal subgroup

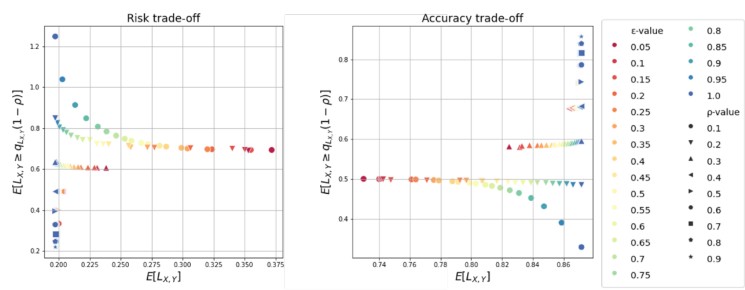

Figure 2: Toy example illustrating the flexibility of R-CVaR objective for hyperparameter $\epsilon \in (0,1]$ and $\rho \in (0,1)$ on synthetic data. FedRCVaR is trained for $\rho = \{0.1,\ldots,0.9\}$ and $\epsilon = \{0.05, 0.1, \ldots, 0.95, 1.0\}$. Different colors describe various $\epsilon$ values, while the markers define a particular $\rho$ value. We report the CVaR and average risks and accuracies.

robust solution for $\epsilon \approx 0$, also given by BPF, and provide similar solution to the centralized ERM for $\epsilon = 1$. Interestingly, we observe that it also increases the model's fairness and utility performance simultaneously, outperforming the FL baselines in most settings. For particular values of $\epsilon$ and $\rho$ it is also able to recover a solution that achieves client robustness, even though we did not design our objective to explicitly provide it.

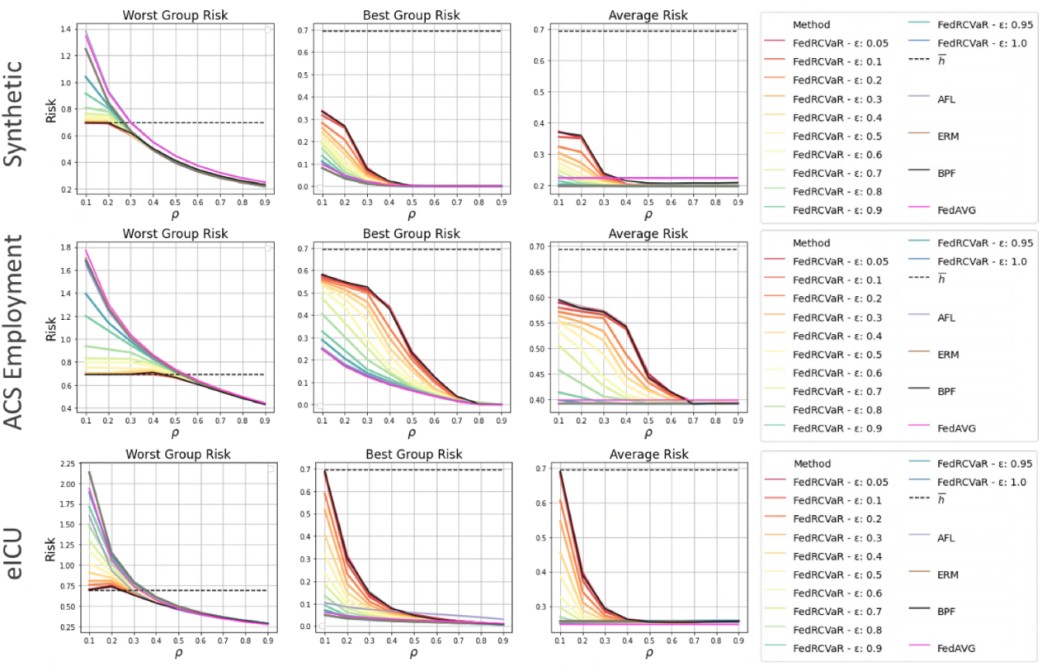

Figure 3: Cross Entropy risks comparison on synthetic, ACS Employment and eICU datasets. FedRCVaR recovers solutions equivalent to centalized machine learning for $\epsilon = \{0.05, 1.0\}$, while improves both utility and accuracy compared to FL baselines in many settings.

## 6  Conclusion

In this work, we address group fairness without access to demographics in FL settings. We design a optimization objective that admits different trade-offs between (sub)group robustness and average utility. We develop an algorithm that efficiently solves the proposed objective. Finally, we perform experiments to illustrate the solutions that can be achieved through the proposed objective and empirically evaluate the efficiency compared with existing works.

## Acknowledgments

UCL authors were supported by Cisco under grant #217462. Duke University authors were partially supported by Cisco, DoD, NSF, and Simons Foundation.

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

# A Experimental Details

## A.1 Synthetic Dataset

We develop a dataset to learn a binary classification task, $Y \in \{0,1\}$, in a federation with two clients. Each client owns features sampled from a truncated normal distributions with means $\{\mu_0, \mu_1\} = \{-1, 1\}$, common variance $\sigma^2 = 1$ and that lie in the intervals $\{(1-, 0.5), (-0.5, 1)\}$, respectively. We consider $p(Y|X) = l\mathbb{1}[x \leq -0.5] + u\mathbb{1}[x \geq 0.5] + m\mathbb{1}[-0.5 < x < 0.5]$, with $\{l, m, u\} = \{0, 0.5\sin\frac{\pi}{2}x + 0.5, 1\}$. We assume that the testing distribution is a uniform $p_{test}(X) = \mathcal{U}(-1, 1)$.

## A.2 Experimental Settings

We preprocess ACS Employment as described in [4] and eICU akin to [20]. eICU dataset requires credentialed access and the procedure for requesting access is described on the dataset's website https://eicu-crd.mit.edu/gettingstarted/access/. For the synthetic and ACS Employment datasets we use a MLP of an hidden layer with size $512$ and for eICU we use logistic regression. We train using Cross Entropy in all cases. FedRCVaR, is trained using batch size equal to the size of client dataset for a single epoch. We use the same options for AFL. FedAvg is trained using batches of sample size $128$ and local epochs $E = 8$. In all scenarios we use model learning rates $\eta_\theta = 0.001$, adversary/threshold learning rate $\eta_c = \eta_{adv} = 0.001$ (where relevant).

For the proposed approach, FedRCVaR, we report the results for group size $\rho = \{0.1, 0.2, 0.3, 0.4, 0.5, 0.6, 0.7, 0.8, 0.9\}$ and trade-off hyperparamenter $\epsilon = \{0.05, 0.1, 0.2, 0.3, 0.4, 0.5, 0.6, 0.7, 0.8, 0.9, 1.0\}$ unless stated otherwise. In the figures we present the mean performance over three runs and in separate splits. The splits are generated using 3-fold cross validation.

## A.3 Implementation & Training Devices

The experiments were conducted in Python using PyTorch. We produce results for BPF [15] using the original code available at github.com/natalialmg/BlindParetoFairness. The experiments were realised using $4\times$ NVIDIA Tesla V100 GPUs.

