# OpenReview forum: "Federated Fairness without Access to Demographics"
_NeurIPS.cc/2022/Workshop/Federated_Learning — FL-NeurIPS 2022 Poster_

### Official Review · Reviewer_hoqA · 2022-10-07
**Strong Results, Slightly Lacking in Clarity**

The paper presents FedRCVaR, an approach to smooth interpolation between sub-group aware/unaware federated learning by means of a hyper-parameter providing a weighted tradeoff. The results are strong and the interpolation between full sub-group awareness and standard federated averaging are clearly visible in the results.

Strengths:
* The results are strong and paint an interesting picture of the capabilities of the proposed method.
* The provided algorithm sheds light on how an implementation of the proposed method would work in practice.
* The method is compared to other prior works in the field on for federated learning with sub-group tradeoffs.

Weaknesses:
* The method is not compared empirically to other federated learning approaches that create or identify sub-groups of users ([1, 2]). Discussion of how the proposed method relates to explicit cluster discovery would be interesting.
* The results are difficult to interpret, as small colored-lines are difficult to distinguish in small formats or black-and-white printing.
* The paper would benefit from more discussion of the results and their implications, as it is currently left to the reader to draw conclusions based on the results shown in Figure 3.

[1] Tang, Xueyang, Song Guo, and Jingcai Guo. "Personalized Federated Learning with Clustered Generalization." arXiv preprint arXiv:2106.13044 (2021).

[2] Silva, A., Metcalf, K., Apostoloff, N., & Theobald, B. J. (2022). Fedembed: Personalized private federated learning. arXiv preprint arXiv:2202.09472.

---

### Official Review · Reviewer_mcpd · 2022-10-18


Summary

This paper explores group fairness in FL where clients are unaware of the demographic group composition in their data. The authors design an optimization objective that admits different trade-offs between (sub)group robustness and average utility and propose an algorithm that solves the objective. Experiments show that the proposed method is effective compared to existing works.


Strength
1. The paper studies an interesting problem of fairness in federated learning.
2. Propose a novel algorithm to alternatively update the model and the quantile parameter to improve the subgroup fairness and average utility.


Weakness
1. It might be better if the authors could improve the fairness problem presentation in section 3.1 and 3.2. Now it introduces the worst-samples performance as the upper tail behavior of the data distribution in eq 3, but it is not clear how it is related to fairness.
2. Convergence analysis of algorithm 1 might further strengthen the submission.

---

### Official Review · Reviewer_5VTt · 2022-10-18
**borderline**

This paper studies the problem of learning a fair model in federated setting without the sensitive groups information. The authors proposed a new objective function for optimization, which interpolates between the vanilla ERM objective, and the expected worst case group performance across the clients.

Pro:
The work is relevant to the workshop theme. Paper is overall well written, with clear theoretical analysis and experimental comparisons.

Cons:
My main concern is about the novelty, and the contribution in the federated learning setting.
(1) It seems that the same interpolated objective is very natural (given that worst case group performance objective has been proposed in prior works). The interpolation is only adding another part which is the vanilla ERM objective.
(2) It is unclear how the federated learning setting is adding difficulty to optimizing the proposed objective function. Due to linearity of the expectations, the main difference seems to just adding an ( n_k / n ) weights to each client k. The contribution of learning such an objective in the federated learning setting compared to prior works in the centralized setting is not very clear to me.

---

### Decision · Program_Chairs · 2022-10-20

Accept (Poster)